# Cell Reprogramming for Regeneration and Repair of the Nervous System

**DOI:** 10.3390/biomedicines10102598

**Published:** 2022-10-17

**Authors:** Isaac H. Clark, Alex Roman, Emily Fellows, Swathi Radha, Susanna R. Var, Zachary Roushdy, Samuel M. Borer, Samantha Johnson, Olivia Chen, Jacob S. Borgida, Aleta Steevens, Anala Shetty, Phoebe Strell, Walter C. Low, Andrew W. Grande

**Affiliations:** 1Department of Biomedical Engineering, Biomedical Engineering Graduate Program, University of Minnesota, Minneapolis, MN 55455, USA; 2Department of Neurosurgery, University of Minnesota, Minneapolis, MN 55455, USA; 3Stem Cell Institute, University of Minnesota, Minneapolis, MN 55455, USA; 4Department of Neuroscience, Graduate Program in Neuroscience, University of Minnesota, Minneapolis, MN 55455, USA; 5Molecular, Cell, Developmental Biology & Genetics Graduate Program, University of Minnesota, Minneapolis, MN 55455, USA; 6Comparative and Molecular Sciences Graduate Program, University of Minnesota, Minneapolis, MN 55455, USA

**Keywords:** neuro-regeneration, cell-reprograming, gene therapies

## Abstract

A persistent barrier to the cure and treatment of neurological diseases is the limited ability of the central and peripheral nervous systems to undergo neuroregeneration and repair. Recent efforts have turned to regeneration of various cell types through cellular reprogramming of native cells as a promising therapy to replenish lost or diminished cell populations in various neurological diseases. This review provides an in-depth analysis of the current viral vectors, genes of interest, and target cellular populations that have been studied, as well as the challenges and future directions of these novel therapies. Furthermore, the mechanisms by which cellular reprogramming could be optimized as treatment in neurological diseases and a review of the most recent cellular reprogramming in vitro and in vivo studies will also be discussed.

## 1. Introduction

### 1.1. Cell Reprogramming

Cell reprogramming is an emerging technology that aims to convert the phenotype of one cell type into another. A classic example is the work of Takahashi and Yamanaka [1] who converted terminally differentiated embryonic and adult mouse fibroblasts into induced pluripotent stem cells (iPSCs)—a cell type that exhibits embryonic stem cell (ESC) properties—through transduction with transcription factors (TFs) typically expressed by ESCs. The iPSCs, in turn, have the potency to differentiate into cells of all lineages. The success of cell reprogramming is dependent on the selection of the preliminary target cell to be reprogrammed, the delivery system that introduces the genes that mediate conversion, the promoters designed to drive gene expression, and the genes that drive reprogramming. Where initial successes with cell reprogramming were limited to conversion in vitro (Figure 1), this technology has now been applied in vivo (Figure 2) with several organ systems to restore nervous system function for various neurological diseases and disorders. This review will highlight recent advances in cell reprogramming for regeneration and repair of the nervous system within the context of several neurological pathologies.

### 1.2. Cell Reprogramming to Generate Neurons

Due to the limited ability of the nervous system to self-repair, reprogramming non-neuronal cells into neurons would provide a substantial benefit to the treatment of several neurological pathologies. While successful neuronal reprogramming as a clinical treatment has not yet been achieved, the hurdles and challenges that emerged in the meantime have brought about surprising new insights to understanding reprogramming [2].

In the application of neuronal replacement therapies, three approaches have emerged to differentiate the current therapeutic efforts: recruitment of neural stem cell (NSC) niches to produce neurons, reprogramming of local glial cells into neurons, and transplantation of fetal progenitor cells. NSC niches are naturally recruited when they are needed, a process known as neurogenesis. This process has been shown to be promoted by the upregulation of proteins such as neuroglobin [3]. Transplantation of fetal progenitor cells, the therapeutic approach closest to a clinical therapy, has been attempted in Parkinson’s disease (PD) patients to varying degrees of success [4,5]. Conversion of local glial cells into neurons has also shown progress with the use of small molecules important for neuron reprogramming and with viral-mediated ectopic expression of pro-neural factors (Figure 3) (Table 1). A recent success with this approach was the conversion of striatal astrocytes into dopaminergic neurons resulting in the recovery of some behavioral symptoms [6]. While each approach has its respective merits, one notable benefit of direct reprogramming is its ability to bypass the lengthy iPSC stage while decreasing the chance of tumorigenesis from latent pluripotent cells [7].

Successful reprogramming of cells into neurons comes with general and cell-specific considerations. In selecting the target cell for reprogramming, significant lineage factors can work to either facilitate or disrupt the reprogramming process. The Waddington Landscape Model [51] characterizes cell specification and determination along its developmental timeline. For cell reprogramming, the model can be used to describe the efficiency of differentiation through the relationship between conversion with how close the target cell is to the differentiated cell of interest, indicating that developmental closeness does serve as a factor for reprogramming success. This is true for the high-efficiency conversion of astrocytes to neurons, though exceptions do exist. While great advances have been made in neuronal reprogramming from immature cells [52], reprogramming mature, developmentally-spaced cells into neurons remains elusive. Cell-specific considerations give additional insight into the difficulty of reprogramming these developmentally-spaced cells into neurons. A potent example is the expression of the RE-1 transcription repressor complex (REST) in non-neuronal cells that functions partly to repress neuronal gene expression [53]. Subsequently, ablating REST expression showed a 90% conversion rate in Neurog2-induced astrocytes [9]. As such, targeting cell-specific machinery to achieve successful cell-to-neuron conversion is a significant factor to consider for reprogramming.

## 2. Genetic Engineering for Cell Reprogramming

### 2.1. Vector and Promoter Design

A critical factor for reprogramming is the precise and intricate genetic engineering strategy employed for the controlled ectopic expression of the proneural gene of interest in the target cell. Pioneer reprogramming studies have successfully employed the Tetracycline-On (Tet-On) inducible gene expression system [54] for neuronal reprogramming. The Tet-On system is a versatile tool that enables Doxycycline-induced regulation of viral gene constructs. Combined with the viral expression regulation under the constitutive *Rosa26 promoter* or the hybrid *CMV-CAG* promoter enhancer combination, this tool was successfully employed to reprogram fibroblasts and microglia to neuronal precursors and mature neurons [6,38,55,56].

With advancements in the field of neuronal reprogramming, the classical genetic strategy of the Cre-LoxP system emerged as a prominent tool to engineer the tailored expression of TFs in target source cells. The bacteriophage protein Cre-recombinase promotes high specificity DNA recombination at sites flanked by short palindromic lox sequences which, in turn, aids in the investigation of gene regulation in the eukaryotic system [57,58,59]. Further refinement of the Cre-loxP recombination system enabled the generation of ligand-inducible recombination, such as tamoxifen, to achieve spatiotemporal manipulation of gene expression in animal models [60,61]. This was successfully employed to drive expression of key proneural TFs for reprogramming, such as NeuroD1 and Dlx2, in mouse models of Alzheimer’s disease [15], Huntington’s disease [24], and ischemic brain injury [17]. Further, a dual-construct design where the primary construct coded for Cre recombinase under the glial fibrillary acidic protein (*GFAP) promoter*, and the secondary construct consisted of the floxed/DIO-FLEX sequences for the TF and fluorescent reporter proteins under a ubiquitous promoter, enabled lineage tracing of the reprogrammed neurons [24].

The revolutionary gene editing system of CRISPR-Cas9 has been successfully employed for astrocyte-to-neuron conversion in vitro as well as in a PD mouse model. The ingeniously modified system of CRISPR-CasRx [62] and CRISPRi [63] allow for precise RNA targeting and manipulation for transcriptional silencing of target genes to achieve astrocyte-to-neuron conversion. In combination with the Cas9 protein and the guide RNA (sgRNA), the CRISPR RNA system has been utilized to initiate repression of the RNA-binding polypyrimidine tract-binding protein (PTBP1) to promote neuronal reprogramming in vivo [29]. Similarly, microRNA gene silencing has also been used to repress PTBP1 expression for the purpose of in vivo reprogramming [30].

For in vivo implementation, the choice of promoter that drives ectopic expression of proneural genes is imperative to achieve high specificity targeting for distinct tissue and cell types. The glial fibrillary acidic protein is an astrocyte intermediate filament protein [64] that is upregulated in reactive astrocytes in response to inflammation, neurodegeneration, and other forms of central nervous system (CNS) injury [65]. Thus, *GFAP promoter*-driven ectopic expression of proneural factors presents a potential dual-pronged approach to reprogramming: a strategy designed for both neuronal replenishment and depletion of the potentially deleterious forms of reactive astrocytes that may contribute to a secondary phase of inflammation which can prolong the CNS injury. The *GFAP promoter* further serves to target astrocytic glial lesions observed in traumatic brain injuries and spinal cord injuries, where there is a need to promote neuronal connections to various other regions through the glial lesion.

### 2.2. Retroviruses and Lentiviruses

Retroviruses and Lentiviruses are RNA viruses that can be engineered to selectively express genes of interest in targeted cells. Retroviruses and Lentiviruses are members of the Retroviridae family of viruses where their differences are in their degrees of complexity; the retrovirus refers to a simple retrovirus whereas the lentivirus is the complex retrovirus [66,67]. Simple retroviral RNA viruses infect dividing cells and propagate their viral profile by integrating the viral genetic material into the host cell genome during mitosis such that, after cell division, the viral genetic material will be expressed in all daughter cells. Lentivirus can infect both non-dividing and dividing cells. Unaltered RNA viruses also have the capability to utilize host cell machinery to replicate and release viruses to infect neighboring cells. For experimental and clinical usage, the viral open reading frames (ORFs) are removed to eliminate viral replicative ability and an additional therapeutic sequence is added to maintain the sequences needed for genome insertion [66].

The simple retroviral genome consists of three major genes—*gag*, *pol*, and *env*—each in their own ORF [67]. The *gag* gene encodes structural glycoproteins. The *pol* gene encodes the enzymes and proteins responsible for replication, cDNA generation, and genomic integration: protease, reverse transcriptase, and integrase. The *env* gene encodes proteins necessary for the surface envelope protein and membrane that enable viral attachment and fusion to host cells. The lentiviral genome contains additional genes that encode transcription and gene expression enhancers—*tat* and *rev*—and accessory proteins—*env, vpr, vif*, and *vpu* [68]. For experimental and clinical purposes, the viral vectors are created through the transfection of transfer plasmids containing non-viral structural and enzymatic genes and the gene of interest under a relevant promoter. Simple retrovirus production requires the transfection of a single plasmid containing all necessary elements, whereas lentivirus production requires the co-transfection of a packaging cell line with several transfer plasmids containing the necessary elements such as to ensure biosafety [67]. The resulting viral particles, packaged with genes of interest under strong promoters, are collected and used to transduce target cells both in vitro and in vivo. For in vitro transduction, the virus-containing medium is added to the sub-confluent target cells where only dividing cells will take up the retroviral vectors. In the dividing cells that take up the retroviral vectors, the viral DNA will integrate into the target cell genome, ultimately resulting in the expression of the gene of interest in the target cells [69]. The daughter cells of these transfected cells will also express the viral gene of interest.

Studies using retroviruses to target dividing cells or lentiviruses to target both non-dividing and dividing cells for reprogramming implement different strategies to alter the tropism of the viral vectors. [70] Such as the use of cell-specific promoters [71], replacement of retroviral envelope proteins [72], and the use of post-transcriptional regulatory elements [73]. An example study of astrocyte-to-neuron reprogramming using retrovirus vectors with a modified retroviral envelope protein used a VSV-G (vesicular stomatitis virus glycoprotein)-pseudotyped retrovirus carrying one of Neurog2 or Dlx2 under a p*CAG* promoter (an internal chicken β-actin promoter with cytomegalovirus enhancer) to selectively target astrocytes [8]. Another study used a human glial fibrillary acidic protein *(hGFAP) promoter* within a lentivirus to control for the selective gene expression in astrocytes [10].

### 2.3. Adeno-Associated Viruses

The Adeno-Associated Virus (AAV) has emerged as a promising gene delivery platform for neuronal reprogramming. First identified as a non-replicating, helper-dependent virus [74], the AAV genome consists of single-stranded DNA (ssDNA). Inverted Terminal Repeats (ITR) essential for viral replication and packaging flanks the ssDNA, with two ORFs encoding the Replication gene (*Rep*), which is required for the AAV replication cycle, and the capsid protein (Cap). The 4.7 kb ssDNA genome of AAV was first characterized to exhibit a site-specific integration at AAVS1 site, a specific locus on chromosome 19 in the host cell genome with limited off-target effects [75,76]. Thus, the AAV gained prominence rapidly as a gene therapy vector [77]. Interestingly, the AAV genome is primarily organized into concatemers in the transduced cells, leading to episomal expression [78]. With the technology of neuronal reprogramming making great strides to achieve cell replacement and consequently, functional benefit in animal models, the next steps entail advancement towards the clinical application for diseases and injuries of the CNS. There is a need to adopt safe, efficient, and reproducible technological platforms for neuronal reprogramming. AAVs, by virtue of being non-pathogenic with low rates of integration into the host genome, have gained approval by the Food and Drug Administration (FDA) as a safe and efficient technology for gene therapy [79]. To date, numerous distinct strains, or serotypes, of the AAVs with diverse cell- and tissue-type specificity have been isolated and characterized [80]. Additionally, chemical modifications of the AAV capsid enables creation of a versatile tool with a wide-range of tissue and cell tropism [81]. Furthermore, the ability of AAV to cross the blood-brain barrier (BBB) to preferentially transduce neurons and glia offers a safe intravenous route for clinical administration in humans [82,83,84,85]. With continual new research, new AAV’s have been produced and will continue to be produced with greater and greater efficiency. For example, AAV-PHP.eB and AAV-MaCPNS1, which have both been shown to successfully transduce neurons via an intravenous injection [85,86].

## 3. In Vitro Cell Reprogramming for Generating Neuronal Cells

### 3.1. Astrocyte to Neuron Reprogramming

Astrocytes constitute a classification of glial cells ranging in subtypes and functions within the CNS. The two main subtypes of astrocytes are protoplasmic and fibrous astrocytes. Protoplasmic astrocytes, primarily found in the gray matter, regulate blood flow through contact with blood vessels and modulate synaptic strength and elimination by direct contact and secreting soluble factors in the tripartite synapse. Fibrous astrocytes, primarily found in the white matter, have been associated with myelinated axons through direct contact with the nodes of Ranvier. During development, astrocytes promote and regulate synapse formation. Adult astrocytes have been shown to play a role in maintaining chemical homeostasis, energy storage, synapse regulation, and response to injury [87]. In response to nervous system injury, astrocytes are recruited to the site of injury for the clearance of molecules, such as amyloid beta and alpha-synuclein, and regulating calcium signaling. Additionally, in response to injury and inflammation, astrocytes adopt a reactive phenotype, increase their proliferative character, and express markers of progenitor cells, making them ideal targets for reprogramming [88].

In vitro astrocyte-to-neuron reprogramming is an experimental approach to creating neurons from existing cultured astrocyte populations. The technique makes use of populations of astrocytes that are susceptible to fate changes in response to exogenous input. Evidence of successful reprogramming is demonstrated through changes in morphology, function, and gene expression. Key studies have identified several small molecules and TFs that can mediate this process, driving specific neuronal subtype fate changes from primary astrocyte cultures. As a result, two main approaches to drive reprogramming in vitro have emerged: viral transduction and small molecule cocktails.

Through retroviral transduction, studies have demonstrated the capability of the neuronal development TF Brn2, to drive astrocyte to glutamatergic neuron conversion [13]. Other reprogramming factors that drive reprogramming with retrovirus transduction include: Ascl1—a bHLH protein implicated in neuronal commitment—to create GABAergic neurons [12], Neurog2—a bHLH protein that specifies neuronal fate—to create GABAergic, glutamatergic, and cortical pyramidal neurons [10,12], and Dlx2—a transcriptional activator that regulates ventral forebrain development—to create GABAergic neurons [10]. Reprogramming cultured astrocytes with a lentivirus vector carrying either Neurog2 or Ascl1 was capable of converting the astrocytes into glutamatergic and GABAergic neurons, respectively [11]. Experiments using AAV vectors loaded with NeuroD1—a bHLH protein and neural TF—successfully reprogrammed cultured astrocytes into functional neurons [16].

The first example of small molecule induced reprogramming identified nine small molecules (LDN193189, SB431542, TTNPB, Tzv, CHIR99021, DAPT, VPA, SAG, and Purmo) capable of reprogramming human astrocytes into vGluT1-expressing functional neurons [21]. These small molecules modulate signaling pathways integral to neurogenesis, including BMP, TGF-β, GSK3, and Shh. This method was further refined when only four of the nine small molecules—LDN193189, SB431542, CHIR99021, DAPT—were capable of reprogramming cultured human fetal astrocytes into functional neurons [22]. Transcriptomics analysis of the induced neurons confirmed a direct transcriptomic shift from astrocytic to neuronal identity over time [89].

The current goals of in vitro astrocyte-to-neuron reprogramming studies look to increase reprogramming efficiency and highlight underlying mechanisms facilitating the astrocyte-to-neuron conversion. Reprogramming efficiency remains a concern due to variability with astrocyte subtypes and region specificity. Different subpopulations of astrocytes have been shown to have varying degrees of susceptibility to specific TF-mediated reprogramming [11]. Additionally, the same induction factor can induce different neuronal subtypes depending on the regional subtype. While the transcriptomic analysis identified changes in signaling pathways in transduced cells, [32] the extent to which these specific signaling pathways drive cell reprogramming remains unclear. Future research is needed to characterize the mechanisms underlying the astrocyte to neuron conversion in vitro and optimize the technology to increase overall efficiency.

### 3.2. Microglia to Neuron Reprogramming

Microglia are considered the macrophages of the CNS and belong to the mononuclear phagocyte system. They populate the embryonic brain and persist throughout adulthood, allowing them to take on distinct physiological states dependent on environmental changes throughout the course of their existence [90]. Microglia are involved in the establishment of the neuronal architecture, controlling neuronal fate and pruning synapses in the developing brain parenchyma [91]. Within the CNS parenchyma, microglia namely interact with two main cell types: astrocytes and neurons, where astrocytes can contribute to microglial activation through the production of pro-inflammatory factors and neurons can alter microglial functions and phenotype by providing specific molecular factors [92]. The nature of the relationship between microglia and neurons makes it a promising candidate for cell conversion studies. Additionally, the microglial population can be rapidly replenished from few surviving microglia [93], making it a suitable candidate for restoring lost neurons by direct conversion without exhausting the microglial pool in the brain.

Direct conversion of astrocytes to neurons has been previously reported; however, this approach has raised some concerns regarding whether astrocytes may be gliotic or dysfunctional. As another option, Matsuda et al., has been the first to endeavor direct microglial conversion to neurons both in vitro and in vivo in mice. Using an unbiased approach in testing out various TFs, NeuroD1 had the highest efficiency of converting microglia to neurons; other microglial TFs that showed sufficient conversion capability include Bhlhe22, Prdm8, and Myt1l. NeuroD1 was expressed using lentiviral expression under the control of the doxycycline inducible tetracycline response element *(TRE) promoter* with a conversion efficiency of about 25–35% [32]. These results, however, have been contested by other research groups, resulting the controversy regarding the ability for microglia to be reprogrammed into neurons. Gao et al. and Rao et al. both attempted in vitro reprogramming of microglia with NeuroD1 via lentivirus and retrovirus. Gao et al. found that microglia were unable to be converted into neurons, potentially due to poor transduction, and Rao et al. found that overexpression of NeuroD1 in microglia actually induced apoptosis [15,94]. As such, direct conversion of microglia to neurons seems a promising approach for cell replacement therapy, but these results may need further verification and the functionality of converted neurons must still be evaluated.

### 3.3. Pericyte to Neuron Reprogramming

Pericytes are mural cells that play a large role in microcirculation. Located in the basement membrane, these cells wrap around the endothelial cells lining the capillaries and venules throughout the body. The many vascular functions of pericytes include regulation of cerebral blood flow, vascular development, angiogenesis, and maintenance of the BBB [95,96]. Pericytes also form part of the neurovascular unit (NVU), a complex unit of cells that dictate the relationship between neurons and the cerebral vasculature in order to meet the metabolic demands of the brain [97]. Located at the interface between the brain parenchyma and the blood vessels, pericytes are paramount to NVU function where they enable communication between these collections of cells and the cerebral vasculature. Pericytes also vary greatly in regional morphology and marker expression. As such, pericytes are termed according to their function and morphology [98]. Pericytes accumulate around a CNS injury locally, making them a good candidate for neurological treatments and neuronal conversion [99].

Extending the spectrum of somatic cell types that are able to give rise to neurons upon forced expression of TFs provides a new approach toward cell-based therapy of neurodegenerative diseases. Karow et al. described the in vitro conversion of pericytes in the adult human brain using Sox2Sox2 and Mash1 (Ascl1) TFs [34]. Cells were isolated from living human patients then subsequently cultured in an incubator for up to two weeks. After enough cells are present, they can be characterized by immunochemistry and labeling to ensure only pericytes are present. After characterization, the cells were transduced with retroviral vectors encoding Sox2Sox2 and Ascl1 using tissue-nonspecific alkaline phosphatase (TN-AP) promoter. These TFs act as the neuronal fate determinants that convert pericytes into induced neurons.

Successful cell conversion has been verified through immunochemistry and electrophysiology, such as patch clamping, to confirm their new identity and viability. The induced neuronal cells converted by this method have been shown to have neuronal electrophysiological properties–such as repetitive action potential firing–demonstrating their ability to integrate into neural networks. The observed conversion rate was 25–30%, and the converted cells did not undergo any cell division during retrovirus-mediated transduction with the neurogenic TFs Sox2Sox2 and Ascl1 [35]. In another study, successful direct reprogramming of adult human brain pericytes into functional induced neurons by Sox2Sox2 and Ascl1 resulted in cells in the productive trajectory transiently acquiring an NSC-like state despite the absence of cell division. Additionally, the transcriptomes were dominated by TF families and non-coding RNAs that play key roles in forebrain GABAergic neurogenesis and glutamatergic subclasses of forebrain neurons [36].

### 3.4. Fibroblast to Neuron Reprogramming

Fibroblasts are located within the interstitial space of organs and are responsible for the production of collagen and extracellular matrix (ECM) materials used to maintain the structure in connective tissue and aid in wound healing to allow injured sites to repair damaged tissue. Active fibroblasts are characterized by the large amount of rough endoplasmic reticulum which helps in their production of ECM components. If the rough endoplasmic reticulum is present in smaller amounts and the cells are relatively smaller in size, they are typically characterized as inactive fibroblasts, or fibrocytes [100].

Fibroblasts have been successfully reprogrammed in vitro into iPSCs through retroviral transduction of TFs KLF4, c-MYC, OCT4OCT4 and SOX2 [1]. Successfully reprogrammed cells were identified as pluripotent when transplanted into nude mice where the iPSCs developed tumors with various tissue types and when injected into mice blastocysts where they were integrated in embryonic development. After the cell has entered the embryonic state, it can then be induced into a neuronal cell via specific proteins, TFs, and environmental cues dependent on the desired neuronal type.

To directly reprogram fibroblasts into neuronal cells, lineage-specific TFs or small molecules can be used in varying combinations to induce the desired cell. Caiazzo et al. directly reprogrammed prenatal rodent, adult rodent, and human fibroblasts into dopaminergic neurons using TFs Ascl1, Nr4a2, and Lmx1a [39]. The neurons were shown to have spontaneous electrical activity as well as the ability to release dopamine. Using seven small molecules, (valproic acid, CHIR99021, Repsox, Forskolin, SP600125, GO6983, Y-27632) Hu et al. demonstrated that human fibroblasts could be directly reprogrammed into neurons in vitro [40]. The resulting induced neurons expressed morphology and gene expression similar to iPSC-derived neurons. Whole cell patch clamping revealed that 88% of the chemically-induced neurons expressed electrophysiological properties such as the ability to fire action potentials and the induction of membrane current.

To test the success of fibroblasts reprogrammed into neurons, researchers typically look for biomarkers specific to either fibroblasts or neurons. Not all cells that undergo TF-mediated transduction will become neurons, so researchers optimized for target cell selection by screening for the best array of biomarkers. After selection of the most appropriate target cells, induced neurons were screened for relevant biological and physiological functions. By the end of the reprogramming and selection process, the cells that remain show very similar properties to native neurons [101]. For example, the expression of neuronal markers such as neural cadherin, MAP2, NEUN, and SYNAPSIN1, neurite outgrowths, and the presence of neurotransmitter receptors GRIN1 (NMDA receptor) and GRIA1 (AMPA receptor). This screening and selection protocol highlights the significance of optimizing all conditions for successful reprogramming in future in vitro studies.

## 4. In Vivo Cell Reprogramming for Neurological Disorders

### 4.1. Stroke

Strokes are neurologic events in which infarction of brain tissue results in focal neurologic deficits. This neurologic event is the leading cause of permanent disability and fifth leading cause of death in the United States [102]. Strokes can be further classified as hemorrhagic, caused by the rupture of cerebral blood vessels, or ischemic, resulting from thrombotic occlusion of cerebral blood vessels. Current treatments for ischemic strokes serve to acutely reestablish cerebral blood flow and include manual removal of the occluding thrombus by thrombectomy or dissolution with tissue plasminogen activator. Treatment of hemorrhagic stroke is more limited, with surgical evacuation of hematomas by craniotomies remaining the mainstay of treatment [103]. While these therapies result in acute improvement in mortality and morbidity, they fail to address the recovery of long-term deficits.

Long-term deficits resulting from stroke are caused by neuronal cell death from critical ischemia or disruption of brain tissue by hematoma after the primary injury. Exacerbating this initial damage is the subsequent secondary neuroinflammatory injury in which activated microglia, astrocytes, and endothelial cells secrete pro-inflammatory cytokines and adhesion molecules [104]. As these cells are abundant after acute stroke, these cells have been preferentially targeted in reprogramming studies to generate new neurons for the recovery of long-term deficits. To successfully reprogram local cells into neurons after stroke, Ascl1, Sox2Sox2, and NeuroD1 have been selected as the proneural TFs of interest. Retroviral transduction with combination of these TFs in a murine stroke model was capable of successfully reprogramming microglia, astrocytes, and oligoprogenitor cells into cells with neuron-like morphology. However, this treatment did not result in a clinical score or infarct volume improvement [25]. A separate study using AAV to introduce NeuroD1 under the *GFAP promoter* ten days post-stroke revealed successful reprogramming of astrocytes into functional neurons with significant improvement in both motor and cognitive functions two months post-treatment [17]. Thus, the choices of delivery platform and proneural gene appear to be influential factors in the success of cellular reprogramming in stroke. For further advancement of reprogramming potency, cellular reprogramming of astrocytes into neurons was enhanced with co-administration of anti-apoptotic agent Bcl2 in a study using a retrovirus to introduce Neurog2 [26]. As such, a combinatorial anti-apoptotic and reprogramming therapeutic approach shows promise in prolonging the lifespan of reprogrammed neurons, and further research is required to assess its functional benefit.

Outside of viral vectors, small molecules have also been evaluated as a candidate delivery approach to facilitate cellular reprogramming in ischemic stroke. Oh et al. utilized an episomal plasmid-based reprogramming technique to generate iPSC-neural precursor cells (NPCs) that were transplanted intracerebrally in a rodent stroke model. Following transplantation, stroke animals showed improvements in behavioral and electrophysiological tests. Twelve weeks post-transplantation, iPSC-NPCs were detected and had been differentiated into both neuronal and glial lineages. These cells were found to promote endogenous brain repair, presumably through increased subventricular zone neurogenesis, and reduce post-stroke inflammation and glial scar formation [43]. Ultimately, both viral- and small molecule-mediated delivery approaches show promise in cellular reprogramming to replenish lost neuronal populations after stroke.

### 4.2. Parkinson’s Disease

In vivo reprogramming presents a particularly powerful cell replacement strategy for neurodegenerative diseases, such as PD. PD is a debilitating movement disorder resulting from the irreversible progressive degeneration of midbrain dopaminergic neurons (mDAs) that project from the substantia nigra pars compacta to the striatum [105,106]. Dopamine, the catecholamine neurotransmitter released by mDAs, is involved in motor function regulation [107]. Due to mDA degeneration in PD patients [108], limited dopamine supply leads to hallmark motor symptoms including postural instability, muscle rigidity, tremors, and strained initiation of voluntary movement [109]. While PD is predominantly idiopathic in nature, multiple lines of evidence have identified genetic *PARK* and *SNCA* genes [110] and environmental factors pesticides [111] as contributing factors. Pioneer treatment strategies include pharmaceutical interventions to alleviate clinical symptoms. Oral or intravenous administration of Levodopa (L-DOPA), a naturally occurring dopamine precursor, has been shown to successfully restore striatal dopamine levels and improve motor deficiencies [112,113], and is a first-line treatment for PD patients. However, prolonged administration of L-DOPA exhibits reduced efficacy over time [114]. Alternatively, striatal transplantation of mDA offers a potent disease-modifying therapeutic strategy by replenishing the extensive mDA population lost at the advanced disease stage. Initial clinical trials report successful survival and integration of transplanted human fetal ventral mesencephalic allografts with corresponding amelioration of clinical symptoms [115]. These approaches however require proper tissue preparation to produce a viable graft and immunotherapy for the cells to be properly accepted after grafting. Building on this approach, in vivo direct reprogramming aims to replenish lost neuronal populations by converting resident supporting cells of the brain to functional dopaminergic neurons in situ. This would remove the requirement for any tissue preparation and any immunotherapy.

Pioneering studies have established great success in reprogramming adult somatic cells to induced dopaminergic neurons (iDA) in vitro using lentivirus to ectopically express mDA fate-specifying proneural TFs. Doxycycline inducible Tet-On based expression of the TFs Ascl1, Brn2, Myt1l along with Lmx1a and FoxA2, driven by the *CMV* promoter, successfully converted human-derived embryonic fibroblasts and adult fibroblasts to iDA. The reprogrammed iDA express characteristic mDA markers such as Tyrosine hydroxylase (TH), Nurr1, aromatic L-amino acid decarboxylase and exhibit functional neuronal electrical activity [38]. Interestingly, a minimal combination of Ascl1, Nurr1, and Lmx1a alone successfully instructed mDA fate in mouse fibroblasts and human fibroblasts derived from healthy individuals and PD patients. Orthotopic transplantation of iDA in mice promoted further development and maturation to functional dopaminergic neurons [39,116]. Niu et al. attempted in vivo reprogramming of adult striatal neurons to functional dopaminergic-like neurons (iDALs). Lentiviral introduction of Sox2Sox2 driven by the *hGFAP promoter* and Nurr1, Lmx1a, Fox2a expressed under a ubiquitous *hPGK promoter* was combined with Valproic acid. iDALs expressed dopaminergic markers such as TH, DAT and exhibited active spontaneous and evoked electrical responses [117]. A pilot study in a non-human primate PD model demonstrated successful integration and function of transplanted iPSC-derived dopaminergic neurons. Cynomolgus monkey-iPSCs were differentiated to dopaminergic neurons in vitro through sequential exposure to key regionalizing cues of retinoic acid, sonic hedgehog signaling, Fgf8a, and Wnt. Transplanted iPSCs-DA neurons in the putamen of the MPTP PD monkey models integrated and matured in vivo and promoted improved motor functions in limb use bias behavioral tests [118]. Using a different approach, Kikuchi et al. reprogrammed and differentiated human iPSCs derived from healthy individuals and PD patients to dopaminergic progenitor cells in vitro using the dual-SMAD inhibition and floor-plate induction protocol. Transplantation of the reprogrammed mDA in the striatum of an immune-suppressed MPTP primate PD model demonstrated successful survival, integration, and maturation of the reprogrammed mDA promoting increased spontaneous movement [119]. Rivetti Di Val Cervo et al. employed a combination of three proneural TFs, NeuroD1, Ascl1, and Lmx1a along with a microRNA—mi218, to induce the dopaminergic fate. For selective targeting of astrocytes, the Tet-On system expressed transgenes under the astrocyte *GFAP promoter*. Additional use of small molecules such as a dual-SMAD inhibitor, Wnt, TGF-b, and Shh midbrain-specific patterning cues demonstrated enhanced conversion efficiency. In vivo reprogrammed iDA in the striatum displayed functional characteristics and contributed to recovery of motor function such as improved axial symmetry and gait as well as diminished drug-induced circling behavior in the 6-hydroxydopamine (6-OHDA) mouse model [6].

Interestingly, emerging studies demonstrated neuron-specific transcriptional programs activated in glial cells following the knockdown of PTBP1. Using the AAV vector, CRISPR-CasRx and gRNA, PTBP1 was ectopically expressed in resident striatal astrocytes under the *GFAP promoter* [30]. The iDA exhibited characteristic molecular and electrical properties and restored the motor and behavioral deficits in the 6-OHDA PD mouse models [29,30]. Remarkably, transient suppression of PTBP1 using the novel antisense oligonucleotide (ASO) technology in striatal astrocytes resulted in successful conversion to dopaminergic neurons. *PTBP1*-ASO injected mice also exhibited rescued behavior in drug-induced tests and touch bias tests for the 6-OHDA model [29]. Thus, in situ reprogramming using a combination of proneural TFs offers a promising therapeutic strategy for long-term functional benefit in PD.

### 4.3. Huntington’s Disease

Huntington’s Disease (HD) is an autosomal dominant disease caused by a mutation in the gene encoding the huntingtin protein (HTT). The mutation causes an abnormally increased number of *CAG* repeats. Individuals affected by HD commonly experience motor impairment, cognitive alterations, and mood disorders. Currently, there is no available cure for HD, but there are experimental drugs in different stages of development that target excitotoxicity, dopamine pathways, mitochondrial dysfunction, and transcriptional dysregulation [120]. Further, advancements in gene editing and cellular reprogramming show promise in improving the quality of life and restoring function in HD patients.

One hallmark of HD is progressive nerve cell degeneration. Therefore, cell reprogramming aiming to replenish neuronal populations in the HD brain is a theoretically beneficial gene therapy approach for patients with HD. In R6/2 and YAC128 HD mouse models, astrocytes were converted into GABAergic neurons through rAAV2/5-mediated ectopic expression of NeuroD1 and Dlx2 [24]; 80% of astrocytes were converted to neurons in R6/2 mice and 30% of astrocytes were converted to neurons in YAC128 mice. Effectiveness of treatments in restoring function is key, as degeneration-based nervous system dysfunction is a key component of the disease. As such, the researchers analyzed behavioral performance in response to the gene therapy treatment where R6/2 mice displayed improved walking distances in an open field test and an extended life span after treatment [24]. A separate study targeting HD fibroblasts in vitro utilized mRNA transfer to reprogram cells through ectopic OCT4OCT4, nanog, klf-4, c-MYC, SOX2, and hTERT expression [41]. The treated cells lost their fibroblast-characteristic gene expression and acquired a hESC profile as displayed by the upregulation in crucial pluripotency genes [41]. Direct conversion does pose a risk as it has been shown that fibroblast induced neurons from HD patients exhibit HD like phenotypes such as mutant HTT (mHTT) aggregates and mHTT-dependent DNA damage [121]. Fortunately, HD patient-derived iPSCs have been previously shown to be genetically corrected to eliminate HD-related pathology [122]. Further research must be done to create safe and effective therapeutic treatments for HD patients.

### 4.4. Alzheimer’s Disease

Alzheimer’s disease (AD) is a neurodegenerative disorder characterized by the accumulation of beta-amyloid plaques and tau tangles and is the leading cause of dementia [123]. Although the cause is not yet understood, genetic risk factors include genetic mutations in *APP*, *PSEN1*, and *PSEN2*, genes of amyloid precursor protein, presenilin 1 and presenilin 2, respectively. Acquired risk factors include hypertension, obesity, type II diabetes, and cerebrovascular diseases [124]. Current treatments can improve clinical symptoms but do not address the underlying pathogenesis of the disease. These treatments include cholinesterase inhibitors and antagonists of NMDA receptors (memantine) [123]. Cellular reprogramming is a novel approach to treating AD through regeneration of cortical neurons.

Mertens et al. and Hu et al. reprogrammed fibroblasts from patients with AD in vitro into neurons via the use of TFs and small molecules [40,125]. Gene expression of these induced neurons showed distinct differences from control patients indicating their potential to model AD. This poses the question of whether cell reprogrammed in vivo will develop the disease or not. Fortunately, some researchers have begun to explore in vivo reprogramming with promising results. Guo et al. successfully reprogrammed cortical astrocytes to glutamatergic and GABAergic neurons in a transgenic mice model with AD (5xFAD) via injection of NeuroD1-GFP retrovirus. To assess whether in vivo reprogramming would result in successful reprogramming of astrocytes into neurons in elderly mice, they injected both seven month and 14 month old mice with the same retrovirus. Remarkably, a higher conversion rate was found in the 14-month-old mice models than in the 7-month-old mice models, perhaps due to the higher number of reactive glial cells in the older animals. Cortical slice recordings showed the NeuroD1-converted neurons were functional and connected to surrounding neurons [15]. Ghasemi-Kasman et al. demonstrated cell reprogramming of hippocampal astrocytes into neurons in streptozotocin (STZ) induced AD mice models through injection of microRNA-302/367 + GFP (miR-302/367) expressing lentiviral particles in the dentate gyrus. The mice were split into four groups: intact, STZ, STZ + valproate (VPA), and STZ + miR + VPA. After a period of time, a short term memory test, the Y maze task, and a spatial reference learning and memory test, the Morris water maze, were performed on the mice. Results of these tests indicated improved working memory and spatial learning and memory in the mice that received the miR-302/367 injection compared to the mice that only received the streptozotocin injection [31]. Therefore, cellular reprogramming as a means to treat AD has demonstrated great potential that necessitates further exploration.

### 4.5. Epilepsy

Epilepsy is characterized by periodic abnormal electrical activity causing seizures and is the most common neurologic disorder, affecting 50 million people worldwide [126]. Epilepsies can be divided into three major categories: (1) idiopathic, in which epilepsy spontaneously arises from mutations and the alteration of basic neuronal regulation; (2) acquired, due to identifiable structural brain lesions; and (3) cryptogenic, in which the cause is undetermined. Current treatment of epilepsy includes antiepileptic drugs, surgery, cell therapy, gene therapy, and brain stimulation with improvement in the control of seizures, however approximately one third of all patients with epilepsy continue to have intractable seizures and experience therapy-related side effects [126]. Current evidence cites the pathogenesis of epilepsy as being related to microglial and astrocyte activation, oxidative stress, reactive oxygen species production, mitochondrial dysfunction, and damage of the BBB [126]. Hypothesized to be the most important contributor to epileptogenesis is the impaired GABAergic function in the brain. It is this impaired function that cellular reprogramming aims to rectify.

In order to correct impaired GABAergic function in the epileptic brain, cellular reprogramming has been used to generate GABAergic interneurons in hopes of increasing the inhibition of electrical activity. Colasante et al. demonstrated successful reprogramming of murine fibroblasts into induced GABAergic interneurons after lentiviral introduction of Foxg1, Sox2, Ascl1, Dlx5, and Lhx6 in combination with anti-apoptotic protein Bcl2. These TFs were induced over different time windows and under different promoters. Sox2 and Foxg1 were kept under the *TetO promoter* and induced for 12–14 days in vitro whereas Ascl1, Dlx5, and Lhx6 were kept under a constitutive *hEF1-alpha promoter*. A critical finding of this study was that Ascl1 alone was less effective at inducing GABAergic neuronal fates than Ascl1 with both Foxg1 and Sox2. The generated GABAergic interneurons survived and matured upon transplantation into the mouse hippocampus and demonstrated to be functionally integrated into host circuitry, inhibiting host granule neuron activity [55]. Similarly, Sun et al. demonstrated that combining microRNA with lentiviral introduction of Ascl1, Dlx2, and Lhx6 enhances the production of GABAergic neurons from human pluripotent stem cells. These reprogrammed cells were demonstrated to receive synaptic currents from host neurons [44]. To evaluate the efficacy of cellular reprogramming in not only generating functional GABAergic interneurons, but also in reducing clinical seizure frequency, Heinrich et al. recently reported successful reprogramming of hippocampal reactive glial cells into GABAergic interneurons using retroviral introduction of both Ascl1 and Dlx2 in a murine model of chronic mesial-temporal lobe epilepsy (MTLE). After long term survival, induced GABAergic interneurons displayed a durable and considerable synaptic integration within endogenous networks and formed inhibitor synapses with appropriate target granule cells. Most importantly, they found reduced occurrence and duration of chronic spontaneous seizures in MTLE mice, mediated by the increase in GABAergic interneurons [27]. Overall, the use of cellular reprogramming to generate inhibitory GABAergic neurons is a promising therapy in the treatment of epilepsy and requires further evaluation.

### 4.6. Spinal Cord Injury

Spinal cord injury (SCI) constitutes a heterogeneous category of nervous system injury where the individual can experience a variety of neurological and functional deficits in response to damage to the spinal cord, the vertebrae, or the surrounding tissue of the spinal cord. The primary injury manifests directly on the spinal cord in the form of compression, contusion, or laceration [127]. The secondary injury consists of the pathologies commonly associated with SCI, which include parenchymal hemorrhaging [128], cell death [129], lesion and glial scar formation [130], and Wallerian degeneration [131]. There are currently no therapeutic approaches to restoring nervous system function after SCI. Instead, individuals who have experienced an SCI undergo physical rehabilitative therapy, which has been shown to acutely improve quality of life but does not significantly improve chronic motor and somatic sensory dysfunction.

Current experimental therapeutic strategies attempt to restore nervous system function by creating a growth-permissive environment in the spinal cord [132,133], overcoming inhibitory signaling through cell transplantation and signaling modification [134,135], improving the intrinsic regenerative capability of neurons and axons [136,137], or implementing a neuronal relay system across the injury site [138]. While some of these strategies are currently in clinical trials, they have yet to result in an approved SCI treatment. With advances in translating cell reprogramming techniques to animal models of CNS injury, cell reprogramming offers a promising approach to restoring the nervous system.

Several recent studies have highlighted the potential of targeting various cells for neuronal induction as an approach to restoring the nervous system after SCI. In particular, NG2-expressing glia and astrocytes have been the candidate cells of interest for in vivo reprogramming in murine models of spinal cord injury. Ectopic expression of SOX2 in NG2+ glial cells induced reprogramming into neurons forming synaptic connections with local neurons. Mice with the SOX2 reprogramming treatment were shown to improve forelimb function in the grid-walking test [139]. Select inhibition of EGFR signaling in NG2-expressing glial progenitors was also able to induce neurogenesis and improve locomotor function as measured by Basso, Beattie, and Bresnahan scoring [37]. NeuroD1 has been studied as a key neurogenic factor capable of inducing astrocyte-to-neuron reprogramming for stab wound and contusive SCIs in both mice and rats [19]. Furthermore, using CRISPRa technology, endogenous expression of Neurog2 and Isl1 in astrocytes was sufficient to promote reprogramming, ultimately enabling the astrocytes to acquire phenotypes associated with motor neurons [23].

Current criticisms of these experimental studies cite insufficient evidence of cell reprogramming, questioning the origin of the proposed ‘reprogrammed cell’ [140]. Future studies may aim to overcome this critique through methods of lineage tracing, transcriptomic analysis, and in vivo imaging. Ultimately, the evidence of functional benefits in rodent models of SCI in response to this gene therapy shows promise for use as a therapeutic treatment for SCI.

### 4.7. Traumatic Brain Injury

Traumatic brain injury (TBI) is caused by some form of physical trauma to the brain, most commonly from falls and motor vehicle accidents, and results in temporary or permanent disruption of normal neurological function. This disruption can often be in the form of memory loss, alteration of mental state, impaired senses and/or impaired motor skills. Similar to stroke, the deficits from TBI result from neuronal cell death and breakdown of the BBB due to both primary external injury and secondary inflammatory injury. Current treatments include cognitive therapy, physical and/or occupational therapy, and decompressive craniectomy [141] with improvement of deficits, however true recovery of the lost neuronal population has not yet been addressed. Cellular reprogramming in TBI aims to replenish this lost population of neurons in order to promote further long-term functional recovery.

Reprogramming efforts in TBI have largely targeted local glial cells, most commonly reactive astrocytes, due to their upregulation during secondary inflammatory injury. To facilitate this, a combination of TFs has been introduced through multiple viral vectors. Retroviral introduction of NeuroD1 under the *GFAP promoter* has successfully reprogrammed astrocytes into functional glutamatergic neurons and NG2 cells into functional glutamatergic/GABAergic neurons [15]. Similarly, retroviral introduction of Sox2 and Ascl1 under a *CAG* promoter, but strikingly also Sox2 alone, induced conversion of NG2 glia into neurons in a murine in vivo stab wound injury model. Notably, lentiviral expression of Sox2 in the non-lesioned cortex failed to convert oligodendroglial and astroglial cells into neurons, suggesting focal reprogramming [142]. This location-specific reprogramming was confirmed by a more recent study that used a Cre-On AAV vector containing Nurr1 and Neurog2 under a *GFAP promoter* to reprogram local reactive astrocytes into neurons post-stab wound injury. The combination of Nurr1 and Neurog2 achieved over 80% reprogramming efficiency with a significant increase in the total number of induced GABAergic interneurons in the injured murine cerebral cortex. While reprogramming was observed in the cerebral cortex, it was not observed in the white matter, suggesting a crucial role of region and layer-specific differences in astrocyte reprogramming [28]. To optimize cellular reprogramming in TBI, Gascon et al. introduced Ascl1 and anti-apoptotic agent Bcl2 through retroviral infection and found greatly improved glial-to-neuron conversion after TBI in vivo. Furthermore, they found that ferroptosis inhibitors potently increased neuronal reprogramming through inhibition of lipid peroxidation occurring during fate conversion [14]. Thus, strategies are in place to prevent reprogrammed cell death and improve reprogramming conversion in glial-to-neuron cellular reprogramming. Outside of direct reprogramming of glial cells into neurons, Gao et al. demonstrated successful reprogramming of reactive glia first into iPSCs through retroviral-mediated expression of TFs OCT4OCT4, Sox2, KLF4, and c-MYC, which eventually differentiated into neurons and glia that filled up the tissue cavity induced by TBI [33]. While the phenotype recovery has not fully been explored in these studies, cellular reprogramming in TBI is a promising therapy for regeneration of functional neurons and long-term functional recovery that necessitates further investigation.

### 4.8. Auditory Disorders

Auditory disorders are commonly caused by exposure to loud noises, genetics, injuries, and aging. The most common type of auditory dysfunction is sensorineural hearing loss which causes a diminished ability to hear faint sounds, understand speech, or perceive any sound clearly. This type of auditory disorder is mainly classified by the loss of hair cells (HCs) and spiral ganglion neurons (SGNs) [143]. SGNs, also known as Primary Auditory Neurons (PANs), are responsible for transmitting electrical signals from the inner ear to the central cochlear nucleus in the brainstem [144]. Since PANs and HCs are post-mitotic, once these cells are lost, mammals cannot regenerate them nor replenish the lost populations which results in a hearing impairment. This degeneration is clinically demonstrated with elevated pure tone audiometric thresholds and decreased word recognition. Impairment of PANs can also lead to auditory neuropathy spectrum disorder, a disorder that affects one’s ability to understand speech despite retaining the ability to hear non-speech sounds clearly. People who suffer from auditory disorders can have a lot of variation in the severity along with the cause of their hearing loss. Therefore, therapeutic interventions for auditory disorders require a versatile approach to restoring auditory function.

Cell reprogramming offers a gene therapy approach with the potential to be implemented in a variety of disorders, including auditory disorders, due to its reliance on endogenous cell populations and the use of TFs correlated to the development of a wide variety of cells. In particular, the use of TFs including Sox2, Atoh1, *Neurog1*, and NeuroD1 have some correlation in neurosensory cell development [145]. Current reprogramming research for auditory disorders targets spiral ganglion non-neuronal cells (SGNNCs) for reprogramming into induced neurons that express markers of PAN cells. SGNNCs are composed mainly of glia, reside within the modiolus, and remain after PAN deterioration [144]. These studies utilize Ascl1 and NeuroD1 in vitro to directly reprogram the SGNNCs to induce neurons. Nishimura et al., demonstrated that the co-culture of both Ascl1 and NeuroD1 can produce induced neurons that share neuron morphology, electrophysiology, and express key neuronal markers [46]. Additionally, a co-culture of induced neurons with respective tissue for cochlear hair cells and cochlear nucleus neurons resulted in the generation of new, PAN-like neurons. Another path of treatment for auditory disorders is for the generation of HC where studies attempt to directly reprogram mammalian somatic cells into functional HCs through the use of TFs Six1, Gfi1, Pou4f3, and Atoh1 (GPA) [146]. A few studies were able to demonstrate that the expression or overexpression of GPA promotes the direct conversion of somatic cells into HC-like cells in vivo [146]. The induced hair cells mimicked hair cells with the presence of polarized espin-rich hair bundle-like protrusions and the regulation of 69% of core genes in hair cells. While the successes with hair cell neuronal induction from non-neuronal cells gives promise to the viability of this gene therapy as an approach for auditory disorders, the extent of functional benefit has yet to be explored.

### 4.9. Visual Disorders

Visual disorders have a broad range of causes including blindness, refractive error, age-related macular degeneration, cataracts, glaucoma, and retinitis pigmentosa (RP). Current treatments for visual disorders consist of vitamin supplements, medicated eye drops, laser treatment, and surgical procedures such as corneal transplants [147]. Corneal endothelial cells play key roles in protecting the retina, delivering oxygen and nutrients, and detecting pathogens, making their presence crucial to cure visual disorders [148]. Cell reprogramming has shown promise in repairing damaged rod photoreceptors, restoring tissue function, and generating corneal endothelial cells.

AAV-mediated reprogramming of rod photoreceptors in a mouse model of RP displayed the capability to convert rod photoreceptors into cone photoreceptors by selective targeting of master photoreceptor TFs, Nrl or Nr2e3. The Rd10 mice were administered the AAV9-based gene therapy to activate rod-specific gene transcription, ultimately restoring tissue function [50]. Electroretinography revealed all treated mice had improved b-wave values, suggesting greater cone function and increased preserved outer nuclear layer thickness [50]. Within the context of ischemic injury-based visual disorders, NeuroD1 was also cited as able to convert astrocytes into neurons that acquired cortical neuron profiles that were capable of integrating into cortical circuitry. These reprogrammed cells responded to visual stimuli and eventually acquired orientation selectivity [18]. In addition to viral-mediated reprogramming, small molecules have also been shown to be an effective strategy for cell reprogramming treatment. Small molecule modulators of Nr2e3, like photoregulin1 (PR1), were used to ease the progression of photoreceptor degeneration in mouse models of a dominant type of RP. This study concluded that PR1 was successful in preventing photoreceptor degeneration and could be an effective option for treatment of dominant forms of RP [149]. Future studies will be required to characterize the functional benefit to the visual system in vivo to determine the efficacy of cell reprogramming as a therapeutic approach to restoring visual function.

### 4.10. Multiple Sclerosis

Multiple sclerosis (MS) is an inflammatory disease of the central nervous system in which the immune system attacks the myelin sheath of neurons, which could result in impaired vision, tremors, numbness, and fatigue. Most cases of MS have a relapsing-remitting disease course, in which patients experience a period of worsening symptoms that can last from days to weeks which eventually improves either partially or completely during remission. While there is no cure, current treatments involve relapse therapy, disease-modifying therapies, and symptom management [150]. Cellular reprogramming is a promising approach to improving treatment for multiple sclerosis through the regeneration of neuronal myelin sheaths. Currently, cellular reprogramming efforts have focused on the use of *Sox10* to reprogram cells into oligodendrocytes for remyelination.

As a TF for reprogramming of oligodendrocytes, *Sox10* has been shown to be a major regulator in oligodendrocyte myelination and activation of myelination genes [151]. Astrocytes have been a target cell for reprogramming efforts due to the similarity in origins and epigenetic states of oligodendrocytes. Khanghahi et al. recently demonstrated successful in vivo reprogramming of astrocytes into oligodendrocyte progenitor-like cells in a murine demyelination model through the lentiviral introduction of *Sox10*. *Sox10*-GFP expressing viral particles were injected into cuprizone-induced demyelination mice brains. At 3 weeks post-injection, the majority of GFP-expressing cells in animals which received control vectors were astrocytes as compared to animals which received the *Sox10*-GFP vector where the main population of GFP-expressing cells were positive for oligodendrocyte lineage markers. Astrocytes were also extracted from mouse pups to be transduced in vitro and retransplanted into demyelinated brains for later fate mapping. At 3 weeks post-transplantation, astrocytes showed oligodendrocyte progenitor and mature oligodendrocyte markers. This study demonstrated the feasibility of reprogramming astrocytes into oligodendrocyte-like cells in vivo using a single TF, *Sox10*s [20]. This method has promise for myelin repair in multiple sclerosis patients, however more research is needed to further improve conversion rates and deepen the understanding of this topic.

### 4.11. Aging of the Nervous System

Aging of the nervous system causes a decline in innate tissue regeneration which affects the body in many different ways. Aging of the CNS results in a loss of grey and white matter which leads to cognitive decline. On the other hand, the aging of the autonomic nervous system (ANS) has been associated with structural and functional effects. These changes are mostly seen in ganglia and ANS-controlled functions including heart rate, blood pressure, and temperature. ANS healthy aging is often associated with an elevation of basal sympathetic nervous activity along with the reduction in parasympathetic nervous activity [152].

The decline in tissue regeneration causes a loss of function of adult stem cell and progenitor cell populations. One type of deterioration is the limited regenerative capacity of the CNS multipotent stem cells known as oligodendrocyte progenitor cells (OPCs) [153]. OPCs are the cells responsible for generating new myelin-producing oligodendrocytes for the purpose of remyelination. Remyelination serves to maintain adult CNS function, especially after events and diseases that result in a loss of myelin; the innate capability to remyelinate decreases with age. Preliminary studies introduce the possibility of generating oligodendrocytes from human iPSCs (hiPSCs). These studies are promising but are still very preliminary due to the lack of primary samples and lack of efficient protocols [42]. Garcia-Leon et al. were able to demonstrate that the overexpression of *Sox10* can generate surface antigen O4-positive (O4+) and myelin basic protein-positive oligodendrocytes from hiPSCs in 22 days. These *Sox10*-induced O4+ populations resemble human oligodendrocytes at the transcriptome level and can myelinate neurons in vivo. This was then translated to the use of CRISPR-Cas9 to directly reprogram fibroblasts to OPCs [45]. They designed a non-viral system to enable stable delivery of pools of Synthetic TFs capable of transcriptional activating three key oligodendrocyte lineage master regulatory genes (*Sox10*, *Olig2*, and *Nkx6-2*). This method could enhance NSC differentiation and initiate mouse embryonic fibroblast direct reprogramming toward oligodendrocyte progenitor-like cells.

### 4.12. Gliomas

Gliomas are one of the most common types of primary brain tumors and include astrocytomas, oligodendrogliomas, and ependymomas. Malignant gliomas, such as anaplastic astrocytomas, anaplastic oligodendroglioma, anaplastic oligoastrocytoma, anaplastic ependymomas, and glioblastoma multiforme (GBM) are classified by World Health Organization grade (grades I–IV) [154]. GBM is the most common malignant primary brain tumor, making up approximately 45% of all gliomas, and has a five-year survival rate of 5% [155]. Current treatment consists of surgical resection followed by external beam radiation and concurrent temozolomide maintenance chemotherapy. Despite these treatments, the median survival for patients with newly diagnosed GBM is only 12–18 months [154]. The persistence of gliomas despite maximal modern medical therapies can be attributed to the highly invasive and proliferative nature of glioma cells. Past therapies have attempted to inhibit proliferation of glioma cells through gene transfer, and while overexpression of TFs P53 [156], Pten [157], and Pax6 [158] did affect glioma growth, glioma cells maintained a proliferative state. Thus, current cellular reprogramming strategies have aimed to induce differentiation of glioma stem cells to inhibit their proliferation.

Glioma cells have been successfully reprogrammed into neurons using both viral vectors and small molecules. Zhao et al. demonstrated successful conversion of human glioma cells into functional neurons as well as resulting inhibition of glioma cell proliferation through lentiviral introduction of Ascl1, Neurog2, and Brn2 [48]. Single TFs have also been successful in reprogramming glioma cells into neurons, with lentiviral introduction of Neurog2 demonstrating successful reprogramming of human glioma cells into neuron-like cells and decreased glioma growth with improved survival of tumor-bearing mice [49]. Similarly, Cheng et al. found that lentiviral introduction of Ascl1 reprogrammed glioma cells into terminally differentiated neurons and inhibited the proliferation of glioma cells [47]. Interestingly, they also assessed Neurog2 where they identified a lower conversion rate as compared to Ascl1. Thus, gene-of-choice appears instrumental in successful reprogramming of glioma cells into neurons using viral vectors. When evaluating viral vectors to treat gliomas, more work needs to be done to increase conversion efficiency and glioma targeting specificity of viral transduction using glioma cell type-specific promoters.

In addition to viral vectors, small molecules have been used to reprogram human GBM cells into terminally differentiated neurons using a small molecule cocktail consisting of forskolin, ISX9, CHIR99021 I-BET 151, and DAPT over 13 days [40]. These factors serve to induce neuronal differentiation in various ways: Forskolin, a cAMP agonist, acts as a chemical substitute for OCT4OCT4, one of the Yamanaka factors that maintains pluripotency in stem cells; ISX9 induces neuronal differentiation through Mef2; CHIR99021 inhibits glycogen synthase kinase 3 and induces neural development from pluripotent stem cells; and I-BET 151 inhibits the BET family of proteins and promotes neuronal differentiation of NSCs. Through genetic analysis it was found that the chemical cocktail upregulated Neurog2, Ascl1, Brn2, and MAP2 which resulted in successful neuronal reprogramming. Interestingly, the targeted GBM cells displayed decreased viability and lacked the ability to form high numbers of tumor-like spheroids, contributing to the goal of glioma cell inhibition. Overall, cellular reprogramming in gliomas seems promising in inhibiting the invasive and proliferative nature of glioma cells through differentiation of the glioma cells.

## 5. Challenges

### 5.1. In Vitro Challenges

Reprogramming efficiency remains the most significant challenge facing in vitro cell reprogramming. Of the many factors that influence cell reprogramming, including the choice of target cell, reprogramming factor, and the means of reprogramming factor delivery, the extent to which each influences the ability to induce a change of cell fate remains to be characterized. In particular, studies aiming to reprogram the same target cell using the same reprogramming factor have cited differences in resultant cell subtype without an understanding for what influenced the inconsistencies. Further, cell fate is influenced by a myriad of both intrinsic and extrinsic fate determinant signals. As such, reprogramming established cells with only a single reprogramming factor has given rise to questions of transcriptional mechanistic understanding. In vitro approach to cell reprogramming offers a means to addressing this concern through cell and transcriptional sequencing analyses.

In translating the reprogramming techniques to animal models, the impact of the microenvironment is a consideration not yet made in in vitro studies. Researchers have postulated that the microenvironment—soluble factors, neighboring cells, secreted inflammatory molecules, etc.—may contribute to the efficiency of reprogramming [51]. The extent to which cell-cell interactions impact cell reprogramming capability have yet to be explored in an in vitro model. Future in vitro reprogramming studies should model physiologically-relevant microenvironmental factors to determine reprogramming efficiency within the context of translating the technique to animal models.

### 5.2. In Vivo Reprogramming vs. Neuroprotection

One issue with reprogramming cells in vivo is the ability to be certain regarding the origin and current identity of the cells in question. In viral treatments, transduced cells are typically distinguished from other endogenous cells by the expression of some kind of biomarker. If the biomarker is present in a neuronal cell, it may be correctly or falsely assumed that the cell is a reprogrammed neuron that originated from some other cell type. This has led to recent controversies in the ability for astrocytes to be reprogrammed in vivo [140].

In general, transduction is limited by the use of a promoter. While cell specific promoters are generally only active in specific cell types, they are still present in all other cells and are not even explicitly needed for the transcription of DNA. Thus, even with a cell-specific promoter, other cell types may be transduced. While this is not a significant issue if the specificity of a promoter is sufficiently high, many factors can alter the specificity of a virus. Wang et al. demonstrated that the serotype of AAV, time post viral injection, and even additional TFs used can shift the specificity of a virus away from the promoter used [140].

While the type of promoter and virus can greatly affect the cells targeted, the concentration or titer of the viral injection can also cause misleading results [159]. Viral loads that are too high can cause leakage of the transcription factors from transduced cells. That leakage can then be taken up by other cells, confounding the results and potentially causing un-desired effects in non-target cells.

Due to the uncertainty regarding these marked cells’ original identity, some beneficial effects seen from attempts to reprogram in vivo may be due to neuroprotection rather than reprogramming. Neuroprotection is a process by which endogenous neurons receive increased vitality where normally they may have died. Factors, such as NeuroD1, that encourage reprogramming when expressed in non-neuronal cells can have neuroprotective effects if expressed in endogenous neurons [160]. As such it is often difficult to determine if a treatment is successful due to reprogramming or neuroprotection. It is important that research in this field take additional steps such as lineage tracing or live cell tracking to help identify the true nature of in vivo reprogramming.

### 5.3. In Vivo Viability

In vivo reprogramming is considered one of the more optimal approaches for therapeutic treatments. In contrast to cell transplantation where there is risk of tumorigenesis and of being targeted by the body’s natural immune response, cell reprogramming offers a unique approach that makes use of endogenous cell populations to potentially improve nervous system dysfunctions. These endogenous cells are naturally distributed throughout the body in optimal ways that cannot be achieved through an injection. This process, however, has unique challenges that need to be studied further.

One challenge of in vivo reprogramming is the presence of the target cell type in the target region. If the target cell type is not present, treatment will not work in that region. Cells from other regions may be targeted, but they may or may not be able to migrate to the target region. Additionally, the cells targeted for reprogramming must be replenished in some manner afterwards. Fortunately, cell types such as astrocytes have been shown to proliferate to replenish themselves after reprogramming [161]. It is also vital that the induced neurons both project to the correct targets and are able to create functioning circuits for successful brain repair. If the reprogrammed cells do not form correctly, is it unknown whether the brain’s natural neuroplasticity would be able to account for this or not. Long-term survival of converted cells must be assessed to determine longevity and viability of the induced neurons.

The time point chosen for reprogramming is also an interesting factor that is greatly dependent on the disease being treated. In PD models for example, reprogramming focuses on the acute disease phase which varies between two weeks to one month after the induction of neurotoxin-based PD model in mice. Further investigation is required to address the efficacy of reprogramming at an advanced stage of the disease or in a chronic PD model to better represent the clinical needs. This is true for many other diseases in which patients could be in a number of different stages. Additionally, it has yet to be established whether the converted neurons still could develop the same disease that was present previously. In the case of inherited diseases such as Huntington’s disease, it is possible that reprogrammed cells could develop m*HTT* inclusion and degenerate considering that reprogramming makes use of endogenous cells. Future studies must consider the factors of location, cell population, cell functionality, and timing if they are to further the understanding of this process as a potential treatment.

## 6. Conclusions

Cellular reprogramming has great potential to treat neurological diseases. With an increasing number of methods available to promote reprogramming, each with their own niche advantages, cellular reprogramming is emerging as a versatile approach aimed at restoring nervous system function. There are different methods available to reprogram a wide range of cell targets with select transcription factors to allow for high specificity and targeted reprogramming in vitro and in vivo. Several studies have highlighted the efficacy of this reprogramming framework in a variety of animal models, showing functional reprogrammed cells capable of inserting into existing circuits for a functional benefit. While many challenges exist for this field, particularly with conflicting reports of successful reprogramming, new techniques are being developed to further elucidate the mechanisms and efficacy of this approach. With more definitive results on the efficacy, safety, and functional benefits of these treatments for several models of neurological disorders, significant preclinical and clinical trials may be on the horizon.

## Figures and Tables

**Figure 1 biomedicines-10-02598-f001:**
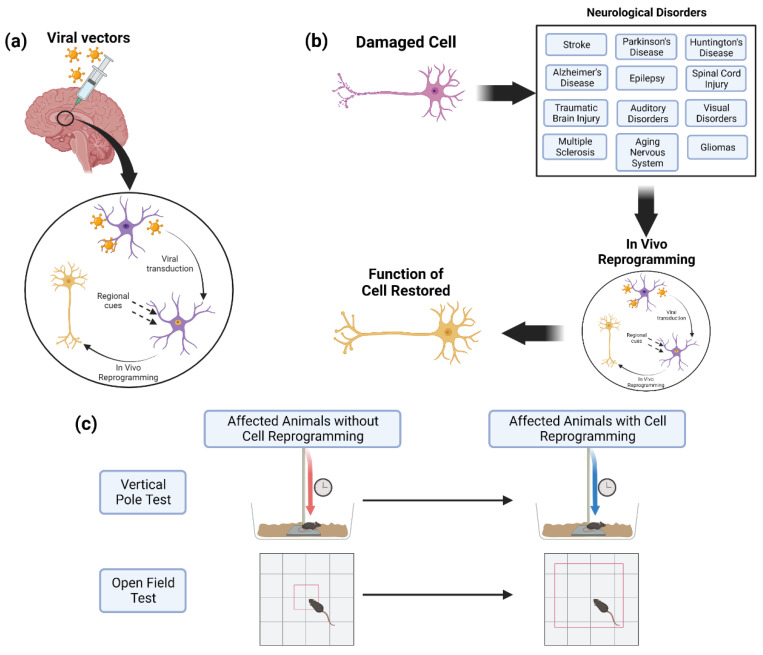
In vivo cell reprogramming. Current research into cell reprogramming has focused on translating the approach to in vivo models of neurological disorders to determine the extent of nervous system repair. (**a**) Reprogramming factors are commonly administered to target cell populations through viral transduction. Viral vectors loaded with the reprogramming transcription factor controlled under a target cell-specific promoter is injected into the site of reprogramming interest to induce in vivo cell reprogramming in transduced target cells. (**b**) One significant benefit to cell reprogramming as a therapeutic approach is its adaptability toward targeting various neurological disorders for nervous system restoration. Cell damage and loss is a common pathology across several neurological disorders where in vivo reprogramming would serve to replenish lost cell populations and recover cell function. (**c**) Functional recovery remains a significant outcome measure of successful in vivo reprogramming in animal models of neurological disorders where animals would hypothetically display functional improvements in behavioral assays.

**Figure 2 biomedicines-10-02598-f002:**
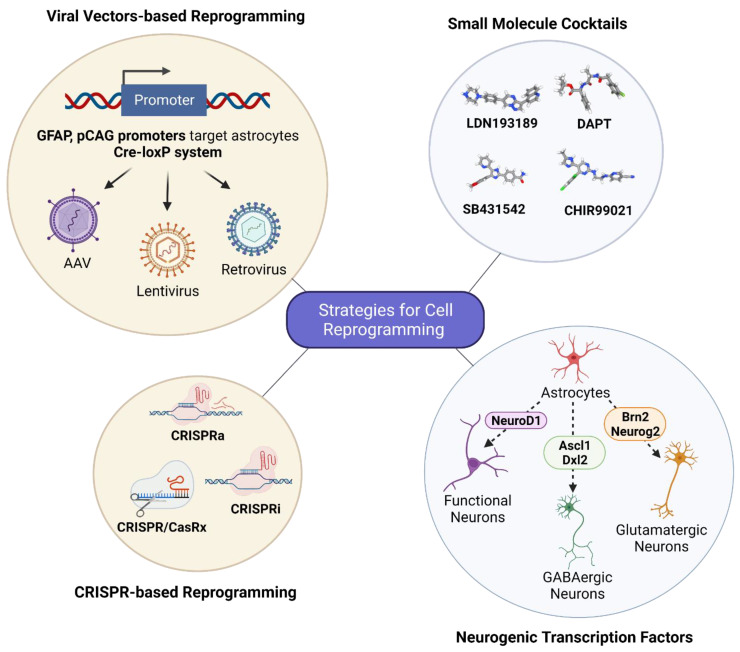
Current strategies to induce cell reprogramming. Methodologies used to reprogram cells in vitro and in vivo include the use of viral transduction, small molecule cocktails, or CRISPR-based gene editing to force expression of neurogenic transcription factors in the target cell populations. Viral-based reprogramming commonly makes use of cell-specific promoters and Cre-lox technology to target specific cell populations for reprogramming. Small molecule cocktails are engineered to upregulate cell signaling pathways involved in neurodifferentiation. CRISPR technology has been used to target cells for specific expression of transcription factors. These transcription factors—such as NeuroD1, Ascl1, Dxl2, Brn2, and Neurog2—are selected as key factors involved in cells differentiating into neurons.

**Figure 3 biomedicines-10-02598-f003:**
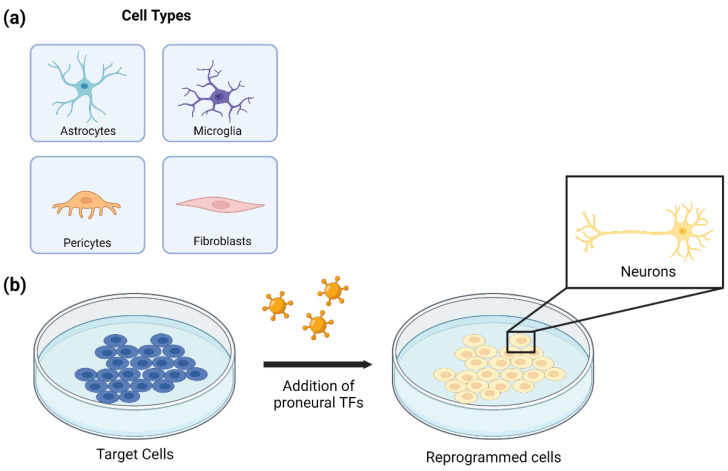
In vitro cell reprogramming. Several cell populations have been identified as targets capable of cell fate reprogramming when cultured with specific reprogramming factors. (**a**) Published literature on in vitro cell reprogramming have targeted astrocytes, microglia, pericytes, and fibroblasts, among others, as candidate cell populations targeted for reprogramming. To achieve cell reprogramming, the targeted cells are cultured with particular transcription factors designed to change the cell fate into a cell of interest. (**b**) This process is commonly designed to generate induced neurons. As such, targeted cells are cultured with proneural transcription factors significant to neurodevelopment. Following administration of proneural factors via small molecule cocktail or viral transduction, target cells are selectively reprogrammed into functional neurons and neuron-like cells.

**Table 1 biomedicines-10-02598-t001:** Summary table of reprogramming methods. This table outlines the cell reprogramming methods highlighted throughout this review organized by target cell, transcription factor(s) used to mediate cell reprogramming, and the details surrounding the method of reprogramming.

Summary Table of Reprogramming Models and Methods			
Cell Type	Factor(s)	Model	Transduction Details
Astrocyte	Neurog2	In vitro	Retroviral-mediated reprogramming [8] enhanced by REST deletion [9] Lentiviral-mediated reprogramming [10,11] Plasmid transfection-mediated reprogramming [12]
	Dlx2	In vitro	Retroviral-mediated reprogramming [8]
	Brn2	In vitro	Retroviral-mediated reprogramming [13]
	Ascl1	In vitro	Lentiviral-mediated reprogramming [11] Plasmid transfection-mediated reprogramming [12] Retroviral-mediated reprogramming enhanced with Bcl-2 expression [14]
	NeuroD1	In vitro Alzheimer’s disease: mouse Ischemic stroke: mouse Spinal cord injury: mouse, rat Non-injured brain: mouse	Retroviral-mediated reprogramming [15] AAV-mediated reprogramming [16] Retroviral-mediated reprogramming [15] AAV-mediated reprogramming [17,18] AAV-mediated reprogramming [19] AAV-mediated reprogramming [16]
	*Sox10*	Multiple sclerosis demyelination: mouse	Plasmid transfection-mediated, *Sox10*-based reprogramming [20]
	Small Molecule and Combinatorial Approaches	In vitro Huntington’s disease: mouse Ischemic stroke: mouse Epilepsy: mouse Traumatic brain injury: mouse Non-injured spinal cord: mouse	Small molecule-mediated reprogramming with nine-molecule cocktail [21] and four-molecule cocktail [22] Combinatorial Ngn2 and Isl1 CRISPRa-mediated reprogramming [23] Combinatorial NeuroD1 and Dlx2 AAV-mediated reprogramming [24] Combinatorial Ascl1, Sox2, NeuroD1 retroviral-mediated reprogramming [25] Combinatorial Ngn2 and Bcl2 retroviral-mediated reprogramming [26] Combinatorial Ascl1 and Dlx2 retroviral-mediated reprogramming [27] Combinatorial Nurr1 and Ngn2 AAV-mediated reprogramming [28] Combinatorial Ngn2 and Isl1 CRISPRa-mediated reprogramming [23]
	Other	Parkinson’s disease: mouse Alzheimer’s disease: mouse	CRISPR-CasRx [29] and Lentiviral [30] based, PTBP1 repression-mediated reprogramming MicroRNA-302/367-mediated reprograming [31]
Microglia	NeuroD1	In vitro Non-injured brain: mouse	Lentiviral-mediated reprogramming [32] Lentiviral-mediated reprogramming [32]
	Combinatorial Approaches	Ischemic stroke: mouse Traumatic brain injury: mouse	Combinatorial Ascl1, Sox2, NeuroD1 retroviral-mediated reprogramming [25] Combinatorial OCT4OCT4, KLF4, Sox2, and c-MYC retroviral-mediated reprogramming [33]
Pericyte	Combinatorial Approach	In vitro	Combinatorial Sox2 and Ascl1 retroviral-mediated reprogramming [34,35,36]
NG2+ Glia	NeuroD1	Alzheimer’s disease: mouse	Retroviral-mediated reprogramming [15]
	Other	Spinal cord injury: mouse	EGFR inhibition-mediated reprogramming [37]
Fibroblast	Small Molecule and Combinatorial Approaches	In vitro	Combinatorial *Oct3/4,* Sox2, c-MYC, and KLF4 retroviral-mediated reprogramming [1] Combinatorial Ascl1, Brn2, Myt1l, Lmx1a, and FoxA2 lentiviral-mediated reprogramming [38] Combinatorial Ascl1, Nurr1, and Lmx1 lentiviral-mediated reprogramming [39] Small molecule-mediated reprogramming with seven-molecule cocktail [40] Combinatorial OCT4OCT4, Nanog, KLF4, c-MYC, Sox2Sox2, and hTERT mRNA plasmid transfection-mediated reprogramming [41]
iPSC	Sox10	In vitro	Lentiviral-mediated Sox10-based reprogramming [42]
	Small Molecule and Combinatorial Approaches	In vitro	Small molecule-mediated reprogramming with dual SMAD inhibitors [43] Combinatorial Ascl1, Lhx6, Dlx2, miR-9/9*-124 lentiviral-mediated reprogramming [44] Combinatorial Sox10, Olig2, and Nkx6-2 CRISPR-Cas9-mediated reprogramming [45]
Other	Ascl1	In vitro	Plasmid DNA transfection-mediated reprogramming of cochlear non-sensory epithelial cells [46] Lentiviral-mediated reprogramming of glioma cells [47]
	Combinatorial Approaches	In vitro Glioma brain tumor: mouse	Combinatorial Ascl1, Brn2, with Ngn2 [48] or Ngn2 with Sox11 [49] lentiviral-mediated reprogramming of glioma cells Combinatorial Ngn2 and Sox11 lentiviral-mediated reprogramming of glioma cells [49] Combinatorial Ngn2 and Sox11 lentiviral-mediated reprogramming of glioma cells in mouse brain tumor [49]
	Other	Visual disorder: mouse	CRISPR-Cas9-mediated Nrl repression for red photoreceptor re-programming [50]

## Data Availability

Not applicable. Figures were created with BioRender.com.

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
