# Peer review of "Cell Reprogramming for Regeneration and Repair of the Nervous System"

_biomedicines, 2022, doi:10.3390/biomedicines10102598_

Round 1
Reviewer 1 Report
The review article “Cell reprogramming for regeneration and repair of the nervous system” by Clark and colleagues describes an overview of recent approaches based on cell reprogramming aimed towards biomedical applications for neurological diseases.
The review is clear and well-structured with few points that could be improved to deliver a more significant outcome for a vast audience.
My advises are the following:
1) The paragraph 3.4 would benefit from a better characterization of the reprogrammed cells from a biological and functional point of view.
2) In the paragraph 4.2 it might be helpful to also cite the cons of the use of fetal ventral mesencephalic allografts, in order to explain why direct reprogramming has emerged as a groundbreaking tool in the field of regenerative medicine.
3) In the paragraph 4.5 and 4.6 there is no information about phenotype recovery after transplantation of the induced cells.
4) The paragraph 4.11 is mainly focused on oligodendrocyte degeneration, but it should have a broader approach (i.e. including altered microglia behaviour).
5) The authors should adjust the figure order to mention Figure 1 as the first mentioned.
6) Gene nomenclature should be adjusted (i.e. Ascl1 instead of Ascl1).
7) Abbreviation consistency should be checked throughout the text (i.e. PD is not used sometimes).
8) Some references should be added. Here are some suggestions:
Parkinson’s disease
Dell'Anno MT, Caiazzo M, Leo D, Dvoretskova E, Medrihan L, Colasante G, Giannelli S, Theka I, Russo G, Mus L, Pezzoli G, Gainetdinov RR, Benfenati F, Taverna S, Dityatev A, Broccoli V. Remote control of induced dopaminergic neurons in parkinsonian rats. J Clin Invest. 2014 Jul;124(7):3215-29. doi: 10.1172/JCI74664. Epub 2014 Jun 17. PMID: 24937431; PMCID: PMC4071410.
Alzheimer’s disease
Mertens J, Herdy JR, Traxler L, Schafer ST, Schlachetzki JCM, Böhnke L, Reid DA, Lee H, Zangwill D, Fernandes DP, Agarwal RK, Lucciola R, Zhou-Yang L, Karbacher L, Edenhofer F, Stern S, Horvath S, Paquola ACM, Glass CK, Yuan SH, Ku M, Szücs A, Goldstein LSB, Galasko D, Gage FH. Age-dependent instability of mature neuronal fate in induced neurons from Alzheimer's patients. Cell Stem Cell. 2021 Sep 2;28(9):1533-1548.e6. doi: 10.1016/j.stem.2021.04.004. Epub 2021 Apr 27. PMID: 33910058; PMCID: PMC8423435.
Hu W, Qiu B, Guan W, Wang Q, Wang M, Li W, Gao L, Shen L, Huang Y, Xie G, Zhao H, Jin Y, Tang B, Yu Y, Zhao J, Pei G. Direct Conversion of Normal and Alzheimer's Disease Human Fibroblasts into Neuronal Cells by Small Molecules. Cell Stem Cell. 2015 Aug 6;17(2):204-12. doi: 10.1016/j.stem.2015.07.006. PMID: 26253202.
Huntington’s disease
Victor MB, Richner M, Olsen HE, Lee SW, Monteys AM, Ma C, Huh CJ, Zhang B, Davidson BL, Yang XW, Yoo AS. Striatal neurons directly converted from Huntington's disease patient fibroblasts recapitulate age-associated disease phenotypes. Nat Neurosci. 2018 Mar;21(3):341-352. doi: 10.1038/s41593-018-0075-7. Epub 2018 Feb 5. Erratum in: Nat Neurosci. 2020 Oct;23(10):1307. PMID: 29403030; PMCID: PMC5857213.
Author Response
I have prepared a revised version in which I address your comments in the following ways.
- Brief description of biological and morphological changes observed
- Brief description of challenges to fetal ventral allografts and how cell reprogramming circumvents those changes
- Phenotype recovery has been included or specified as necessary
- Aged microglia have not explicitly been explored in reprogramming from what I can find. As such we will focus on explaining approaches that are currently being researched with reprogramming.
- figures have been adjusted
- genes have been adjusted
- abbreviations have been adjusted
- references have been added and brief explanations of their work have been added where necessary.
Reviewer 2 Report
This is a remarkable manuscript that extensively explains the cell reprogramming approaches for better neuronal cell reprogramming and treatment for neurodegenerative diseases. The figures look also nice.
My suggestions:
1. In the end, I would add a separate chapter on the conclusion, and the future of neuronal reprogramming.
2. A second table would be nice, which summarizes the cell models, and approaches performed in different neurodegenerative diseases (such as AD, PD, MS, etc)
3. Are there similar approaches in spinocerebellar ataxia, prion diseases, or frontotemporal dementia? You may mention it briefly.
Author Response
- A conclusion with the future of cell reprogramming has been added
- The table has been updated to include disease states that those examples have been studied in
- I found iPSCs derived from these disease models but I have been unable to find examples of reprogramming in them